# Algorithm for Computing Approximate Nash Equilibrium in Continuous Games with Application to Continuous Blotto

**Sam Ganzfried**

Ganzfried Research, Miami Beach, FL 33139, USA; sam.ganzfried@gmail.com

**Abstract:** Successful algorithms have been developed for computing Nash equilibrium in a variety of finite game classes. However, solving continuous games—in which the pure strategy space is (potentially uncountably) infinite—is far more challenging. Nonetheless, many real-world domains have continuous action spaces, e.g., where actions refer to an amount of time, money, or other resource that is naturally modeled as being real-valued as opposed to integral. We present a new algorithm for approximating Nash equilibrium strategies in continuous games. In addition to two-player zero-sum games, our algorithm also applies to multiplayer games and games with imperfect information. We experiment with our algorithm on a continuous imperfect-information Blotto game, in which two players distribute resources over multiple battlefields. Blotto games have frequently been used to model national security scenarios and have also been applied to electoral competition and auction theory. Experiments show that our algorithm is able to quickly compute close approximations of Nash equilibrium strategies for this game.

**Keywords:** continuous game; national security; Blotto game; imperfect information

## 1. Introduction

Successful algorithms have been developed for computing approximate Nash equilibrium strategies in a variety of finite game classes, even classes that are challenging from a computational complexity perspective. For example, an algorithm that was recently applied for approximating Nash equilibrium strategies in six-player no-limit Texas hold'em poker defeated strong human professional players [1]. This is an extremely large extensive-form game of imperfect information. Even solving three-player perfect-information strategic-form games is challenging from a theoretical complexity perspective; it is PPAD-hard[1] to compute a Nash equilibrium in two-player general-sum and multiplayer games, and it is widely believed that no efficient algorithms exist [2–4]. Strong algorithms have also been developed for stochastic games, even with multiple players and imperfect information [5]. Stochastic games have potentially infinite duration but a finite number of states and actions.

Continuous games are fundamentally different from finite games in several important ways. The first is that they are not guaranteed to have a Nash equilibrium; Nash's theorem only proved the existence of a Nash equilibrium in finite games [6]. A second challenge is that we may not even be able to represent mixed strategies in continuous games, as they correspond to probability distributions over a potentially (uncountably) infinite pure strategy space. So even if a game has a Nash equilibrium, we may not even be able to represent it, let alone compute it. Equilibrium existence results and algorithms have been developed for certain specialized classes; however, there are still many important game classes for which these results do not hold. Even two-player zero-sum games remain a challenge. For example, the fictitious play algorithm has been proven to converge to Nash equilibrium for finite two-player zero-sum games (and certain classes of multiplayer and nonzero-sum games), but this result does not extend to continuous games [7].

---

[1] PPAD stands for "Polynomial Parity Arguments on Directed graphs".

A strategic-form game consists of a finite set of players $N = \{1, \ldots, n\}$, a finite set of pure strategies $S_i$ for each player $i$, and a real-valued utility for each player for each strategy vector (aka *strategy profile*), $u_i : S_1 \times \ldots \times S_n \to \mathbb{R}$. A two-player game is called *zero sum* if the sum of the payoffs for all strategy profiles equals zero, i.e., $u_1(s_1, s_2) + u_2(s_1, s_2) = 0$ for all $s_1 \in S_1, s_2 \in S_2$.

A *mixed strategy* $\sigma_i$ for player $i$ is a probability distribution over pure strategies, where $\sigma_i(s_{i'})$ is the probability that player $i$ plays $s_{i'} \in S_i$ under $\sigma_i$. Let $\Sigma_i$ denote the full set of mixed strategies for player $i$. A strategy profile $\sigma^* = (\sigma_1^*, \ldots, \sigma_n^*)$ is a *Nash equilibrium* if $u_i(\sigma_i^*, \sigma_{-i}^*) \geq u_i(\sigma_i, \sigma_{-i}^*)$ for all $\sigma_i \in \Sigma_i$ for all $i \in N$, where $\sigma_{-i}^*$ denotes the vector of the components of strategy $\sigma^*$ for all players excluding $i$. It is well known that a Nash equilibrium exists in all finite games [6]. In practice, all that we can hope for in many games is the convergence of iterative algorithms to an approximation of Nash equilibrium. For a given candidate strategy profile $\sigma^*$, define $\epsilon(\sigma^*) = \max_{i \in N} \max_{\sigma_i \in \Sigma_i} \left[ u_i(\sigma_i, \sigma_{-i}^*) - u_i(\sigma_i^*, \sigma_{-i}^*) \right]$. The goal is to compute a strategy profile $\sigma^*$ with as small a value of $\epsilon$ as possible (i.e., $\epsilon = 0$ indicates that $\sigma^*$ comprises an exact Nash equilibrium). We say that a strategy profile $\sigma^*$ with value $\epsilon$ constitutes an *$\epsilon$-equilibrium*. For two-player zero-sum games, there are algorithms with bounds on the value of $\epsilon$ as a function of the number of iterations and game size, and for different variations $\epsilon$ is proven to approach zero in the limit at different worst-case rates (e.g., [8]).

If $\sigma_i^1$ and $\sigma_i^2$ are two mixed strategies for player $i$ and $p \in (0, 1)$, then we can consider mixed strategy $\sigma_i' = p\sigma_i^1 + (1 - p)\sigma_i^2$ in two different ways. The first interpretation, which is the traditional one, is that $\sigma_i'$ is the mixed strategy that plays pure strategy $s_i \in S_i$ with probability $p\sigma_i^1(s_i) + (1 - p)\sigma_i^2(s_i)$. Thus, $\sigma_i'$ can be represented as a single mixed strategy vector of length $|S_i|$. A second interpretation is that $\sigma_i'$ is the mixed strategy that with probability $p$ selects an action by randomizing according to the probability distribution $\sigma^1$, and with probability $1 - p$ selects an action by randomizing according to $\sigma^2$. Using this interpretation implementing $\sigma_i'$ requires storing full strategy vectors for both $\sigma^1$ and $\sigma^2$, though clearly the result would be the same as in the first case.

In extensive-form imperfect-information games, play proceeds down nodes in a *game tree*. At each node $x$, the player function $P(x)$ denotes the player to act at $x$. This player can be from the finite set $N$ or an additional new player called Chance or Nature. Each player's nodes are partitioned into *information sets*, where the player cannot distinguish between the nodes at a given information set. Each player has a finite set of available actions at each of the player's nodes (note that the action sets must be identical at all nodes in the same information set because the player cannot distinguish the nodes). When play arrives at a leaf node in the game tree, a terminal real-valued payoff is obtained for each player according to utility function $u_i$. Nash equilibrium existence and computational complexity results from strategic-form games hold similarly for imperfect-information extensive-form games; e.g., all finite games are guaranteed to have a Nash equilibrium, two-player zero-sum games can be solved in polynomial time, and equilibrium computation for other game classes is PPAD-hard.

Randomized strategies can have two different interpretations in extensive-form games. Note that a *pure strategy* for a player corresponds to a selection of an action for each of that player's information sets. The classic definition of a *mixed strategy* in an extensive-form game is the same as for strategic-form games: a probability distribution over pure strategies. However, in general the number of pure strategies is exponential in the size of the game tree, so a mixed strategy corresponds to a probability vector of exponential size. By contrast, the concept of a *behavioral strategy* in an extensive-form game corresponds to a strategy that assigns a probability distribution over the set of possible actions at each of the player's information sets. Since the number of information sets is linear in the size of the game tree, representing a behavioral strategy requires only storing a probability vector of size that is linear in the size of the game tree. Therefore, it is much preferable to work with behavioral strategies than mixed strategies, and algorithms for extensive-form games generally operate on behavioral strategies. Kuhn's theorem states that in any finite

extensive-form game with perfect recall, for any player and any mixed strategy, there exists a behavioral strategy that induces the same distribution over terminal nodes as the mixed strategy against all opponent strategy profiles [9]. The converse is also true. Thus, mixed strategies are still functionally equivalent to behavioral strategies, despite the increased complexity of representing them.

Continuous games generalize finite strategic-form games to the case of (uncountably) infinite strategy spaces. Many natural games have an uncountable number of actions; for example, games in which strategies correspond to an amount of time, money, or space. One example of a game that has recently been modeled as a continuous game in the AI literature is computational billiards, in which the strategies are vectors of real numbers corresponding to the orientation, location, and velocity at which to hit the ball [10].

**Definition 1.** *A continuous game is a tuple $G = (N, S, U)$ where*

- $N = \{1, 2, 3, \ldots, n\}$ *is the set of players*
- $S = (S_1, \ldots, S_n)$, *where each $S_i$ is a (compact) metric space corresponding to the set of strategies of player i*
- $U = (u_1, \ldots, u_n)$, *where $u_i : S_1 \times \ldots \times S_n \to \mathbb{R}$ is the utility function of player i*

Mixed strategies are the space of Borel probability measures on $S_i$. The existence of a Nash equilibrium for any continuous game with continuous utility functions can be proven using Glicksberg's generalization of the Kakutani fixed point theorem [11]. The result is stated formally in Theorem 1 [12]. In general, there may not be a solution if we allow non-compact strategy spaces or discontinuous utility functions. We can define extensive-form imperfect-information continuous games similarly to that for finite games, with analogous definitions of mixed and behavioral strategies.

**Theorem 1.** *Consider a strategic-form game in which the strategy spaces $S_i$ are nonempty compact subsets of a metric space. If the payoff functions $u_i$ are continuous, there exists a (mixed strategy) Nash equilibrium.*

While this existence result has been around for a long time, there has been very little work on practical algorithms for computing equilibria in continuous games. One interesting class of continuous games for which algorithms have been developed is *separable games* [13]; however, this imposes a significant restriction on the utility functions, and many interesting continuous games are not separable. Additionally, algorithms for computing approximate equilibria have been developed for several other classes of continuous games, including simulation-based games [14], graphical tree-games [15], and continuous poker models [16]. The continuous Blotto game that we consider does not fit in any of these classes, and in fact has discontinuous utility functions, so we cannot apply Theorem 1 or these algorithms.

## 2. Continuous Blotto Game

The Blotto game is a type of two-player zero-sum game in which the players are tasked to simultaneously distribute limited resources over several objects (or battlefields). In the classic version of the game, the player devoting the most resources to a battlefield wins that battlefield and the gain (or payoff) is then equal to the total number of battlefields won. The Blotto game was first proposed and solved by Borel in 1921 [17] and has been frequently applied to national security scenarios. It has also been applied as a metaphor for electoral competition, with two political parties devoting money or resources to attract the support of a fixed number of voters: each voter is a "battlefield" that can be won by one party. The game also finds application in auction theory where bidders must make simultaneous bids [18].

Initial approaches derived analytical solutions for special cases of the general problem. Borel and Ville proposed the first solution for three battlefields [19], and Gross and Wagner

generalized this result for any number of battlefields [20]. However, they assumed that colonels have the same number of troops. Roberson computed optimal strategies of the Blotto games in the continuous version of the problem where all of the battlefields have the same weight, for models with both symmetric and asymmetric budgets [21]. Hart considered the discrete version, again when all battlefields have equal weight, and solved it for certain special cases [22]. It was not until 2016 that the first algorithm was provided to solve the general version of the game. Initially a polynomial-time algorithm that involved solving exponential-sized linear programs was presented [23], which was later improved to a linear program of polynomial size [24]. These polynomial-time algorithms are for the discrete version of the game; however, no general algorithm has been devised for the original continuous Blotto game. As described earlier, there are many challenges present for solving continuous games that do not exist for finite games, even for two-player zero-sum games.

Most of the prior approaches solve perfect-information versions of the game in which all players have public knowledge of the values of the battlefields. Adamo and Matros studied a Blotto game in which players have incomplete information about the other player's resource budgets [25]. Kovenock and Roberson studied a model where the players are subject to incomplete information about the battlefield valuations [26]. In both of these works, all players are equally uninformed about the parameters. Recently some work has provided analytical solutions for certain settings with asymmetric information, in which both players know the values of the battlefields but one player knows their order while the other player only knows a distribution over the possible orders [27,28]. This model is an imperfect-information game in which player 1 must select a strategy without knowing the order, while player 2 can select a different mixed strategy conditional on the actual order. We study and present an algorithm for the asymmetric imperfect-information continuous version of the Blotto game, which is perhaps the most challenging variant. Note that our approach also applies to the perfect information version as well.

A continuous Blotto game is a tuple $G = (N, F, O, p, v, B, S, \delta, u)$:

- Set of players $N = \{1, 2\}$
- Set $F = \{1, 2, \ldots, |F|\}$ of battlefields
- Set of slots $Q = F$
- Set $O$ of outcomes, which is a subset of the set of permutations of elements of $F$, where $o(q)$ denotes the battlefield in slot $q$ for $o \in O, q \in Q$. Let $M = |O|$.
- Probability mass function $p$ with $p(o)$ for each $o \in O$
- Positive real value $v_f$ for each battlefield $f \in F$
- Positive real-valued budget $B_i$ for each player $i \in N$
- Pure strategy space of player 1 $S_1$ is $\{(x_q) \in \mathbb{R}^{|Q|} | \sum_q x_q = B_1, x_q \geq 0 \ \forall q \in Q\}$. Let $s_1(q)$ denote the probability of selecting slot $q$ for $s_1 \in S_1$.
- Pure strategy space of player 2 $S_2$ is $\{(x_{o,q}) \in \mathbb{R}^{|O||Q|} | \sum_q x_{o,q} = B_2, x_{o,q} \geq 0 \ \forall o \in O \ \forall q \in Q\}$. Let $s_2(o, q)$ denote the probability of selecting slot $q$ under outcome $o$ for $s_2 \in S_2$.
- $\delta \in \mathbb{R} > 0$
- Utility function $u_1(s_1, s_2) = \sum_o p(o) \sum_q C_1(s_1(q), s_2(o, q))$ for $s_1 \in S_1, s_2 \in S_2$, where

  - $C_1(s_1(q), s_2(o, q)) = v_{o(q)}$ if $s_1(q) \geq s_2(o, q) + \delta$,
  - $C_1(s_1(q), s_2(o, q)) = -v_{o(q)}$ if $s_1(q) \leq s_2(o, q)$,
  - $C_1(s_1(q), s_2(o, q)) = 0$ otherwise

- Utility function $u_2(s_1, s_2) = \sum_o p(o) \sum_q C_2(s_1(q), s_2(o, q))$ for $s_1 \in S_1, s_2 \in S_2$, where

  - $C_2(s_1(q), s_2(o, q)) = -v_{o(q)}$ if $s_1(q) \geq s_2(o, q) + \delta$,
  - $C_2(s_1(q), s_2(o, q)) = v_{o(q)}$ if $s_1(q) \leq s_2(o, q)$,
  - $C_2(s_1(q), s_2(o, q)) = 0$ otherwise

Each player must select a real-valued amount of resources to put on the battlefield in slot $q \in Q$, subject to the constraint that the total does not exceed the player's budget $B_i$.

Player 1 does not know the outcome $o$, which defines the order of the battlefields; they only know that the outcome is $o \in O$ with probability $p(o)$. Player 2 knows the order and is able to condition their strategy on this additional information. For each slot $q$, if player 1 uses an amount of resources $s_1(q)$ that exceeds player 2's amount $s_2(o, q)$ by at least $\delta$, then player 1 "wins" the battlefield $o(q)$ in slot $q$ and receives its value $v_{o(q)}$ (and player 2 receives $-v_{o(q)}$); if $s_2(o, q) \geq s_1(q)$ then player 2 wins $v_{o(q)}$ and player 1 loses $v_{o(q)}$; otherwise, both players get zero. This game is clearly zero sum because player 1 and player 2's payoff sum to zero for each situation.

Note that the utility function is discontinuous: payoffs for a given slot can shift abruptly between $v_{o(q)}$, 0, and $-v_{o(q)}$ with arbitrarily small changes in the strategies. This means that Theorem 1 does not apply, and the game is not necessarily guaranteed to have a Nash equilibrium. The game does also not fall into the specialized classes of games such as separable games for which prior algorithms have been developed. Note that often the Blotto game is presented without the $\delta$ term; typically player 1 wins the battlefield if $s_1(q) > s_2(o, q)$, and player 2 wins if $s_2(o, q) \geq s_1(q)$. We add in the $\delta$ term because our algorithm involves the invocation of an optimization solver, and optimization algorithms typically cannot handle strict inequalities. We can set $\delta$ to a value very close to zero.

## 3. Algorithm

Fictitious play is an iterative algorithm that is proven to converge to Nash equilibrium in two-player zero-sum games (and in certain other game classes), though not in general for multiplayer or non-zero-sum games [7,29]. While it is not guaranteed to converge in multiplayer games, it has been proven that if it does converge, then the average of the strategies played throughout the iterations constitute an equilibrium [30]. Fictitious play has been successfully applied to approximate Nash equilibrium strategies in a three-player poker tournament to a small degree of approximation error [5,31]. More recently, fictitious play has also been used to approximate equilibrium strategies in multiplayer auction [32,33] and national security [34] scenarios. Fictitious play has been demonstrated to outperform another popular iterative algorithm, counterfactual regret minimization, in convergence to equilibrium in a range of multiplayer game classes [35].

In classical fictitious play, each player plays a best response to the average strategies of his opponents thus far. Strategies are initialized arbitrarily (typically they are initialized to be uniformly random). Then each player uses the following rule to obtain the average strategy at time $t$:

$$\sigma_i^t = \left(1 - \frac{1}{t}\right)\sigma_i^{t-1} + \frac{1}{t}\sigma_i'^t,$$

where $\sigma_i'^t$ is a best response of player $i$ to the profile $\sigma_{-i}^{t-1}$ of the other players played at time $t - 1$. The final strategy output after $T$ iterations $\sigma^T$ is the average of the strategies played in the individual iterations (while the best response $\sigma_i'^t$ is the strategy actually played at iteration $t$).

The classical version of fictitious play involves representing two strategies per player; the current strategy $\sigma_i^t$ and the current best response $\sigma_i'^t$. Note that once we compute the next round strategy $\sigma_i^{t+1}$ from $\sigma_i^t$ and $\sigma_i'^{t+1}$, we no longer need to maintain either $\sigma_i^t$ or $\sigma_i'^{t_1}$ in memory. We interpret $\sigma_i^t$ as a single mixed strategy that selects action $s_j$ with probability $\left(1 - \frac{1}{t}\right)\sigma_i^{t-1}(s_j) + \frac{1}{t}\sigma_i'^t(s_j)$.

An alternative, and seemingly nonsensical, way to implement fictitious play would be to separately store each of the pure strategies that are played $\sigma_i'^t$, rather than to explicitly average them at each step. Using this representation, the best response can be computed by selecting the pure strategy that maximizes the average (or sum) of the utilities against $\sigma_{-i}'^0, \ldots, \sigma_{-i}'^{t-1}$. This method of implementing fictitious play seems nonsensical for several reasons. First, it involves picking a strategy that maximizes the sum of utilities against $t$ different opponent strategies as opposed to maximizing the utility against a single strategy. And second, it involves storing $t$ pure strategies for each player, which would require

using significantly more memory than the original approach when $t$ exceeds $|S_i|$. Despite these clear drawbacks, nonetheless it is apparent that this approach is still equivalent to the original approach and results in the same sequence of strategies being played. When the algorithm is applied to an imperfect-information game, we can view it as operating with mixed as opposed to behavioral strategies (in contrast to prior algorithms for solving imperfect-information games). We refer to this new approach as "Redundant fictitious play" due to the fact that it "redundantly" stores all of the strategies played individually instead of storing them as a single mixed strategy. Redundant fictitious play is depicted in Algorithm 1.

---

**Algorithm 1** Redundant fictitious play for two-player games

---

**Inputs**: Number of iterations $T$

Initialize strategy arrays $S_1[T]$, $S_2[T]$
$S_1[0], S_2[0] \leftarrow \text{InitialValues}()$
$v_1^*[0] \leftarrow u_1(S_1[0], S_2[0])$
$v_2^*[0] \leftarrow u_2(S_1[0], S_2[0])$
**for** $t = 1$ to $T$ **do**
$\quad S_1[t] \leftarrow BestResponse1(Mix(S_2, 0, t - 1))$
$\quad S_2[t] \leftarrow BestResponse2(Mix(S_1, 0, t - 1))$
$\quad \epsilon_1[t] \leftarrow u_1(S_1[t], Mix(S_2, 0, t - 1)) - v_1^*[t - 1]$
$\quad \epsilon_2[t] \leftarrow u_2(Mix(S_1, 0, t - 1), S_2[t]) - v_2^*[t - 1]$
$\quad \epsilon[t] \leftarrow \max_i \epsilon_i[t]$
$\quad v_1^*[t] \leftarrow u_1(Mix(S_1, 0, t), Mix(S_2, 0, t))$
$\quad v_2^*[t] \leftarrow u_2(Mix(S_1, 0, t), Mix(S_2, 0, t))$

---

In Algorithm 1, we store $T$ strategies for each player, where $T$ is the total number of iterations. We can initialize strategies arbitrarily for the first iteration (e.g., to uniform random). For all subsequent iterations the strategy $S_i[t]$ is a pure strategy best response to a strategy of the opponent.[2] The notation $Mix(S_i, 0, t - 1)$ refers to the mixed strategy for player $i$ that plays strategy $S_i[u]$ with probability $\frac{1}{t}$, for $0 \leq u \leq t - 1$; that is, it mixes uniformly over the strategies $S_i[0], \ldots, S_i[t - 1]$. The algorithm then computes the game value to player $i$ under the current iteration strategies as well as the *exploitability* of each player (difference between best response payoff and game value). This determines the maximum amount that each player can gain by deviating from the strategies; we can then say that the strategies computed at iteration $t - 1$ constitute an $\epsilon_t$-equilibrium, where $\epsilon_t = \max_i \epsilon_i[t]$.

Now, suppose that $G$ is a continuous game and no longer a finite game. Assuming that we initialize the strategies $S_i[0]$ to be pure strategies, all of the strategies $S_i[t]$ are now pure strategies and the algorithm does not need to represent any mixed strategies. This is very useful, since for continuous games a mixed strategy may be a probability distribution that puts weight on infinitely many pure strategies and cannot be compactly represented. However, pure strategies can typically be represented compactly in continuous games. For example, if the strategy spaces are compact subsets of $\mathbb{R}^n$, then each pure strategy corresponds to a vector of $n$ real numbers, which can be easily represented assuming that $n$ is not too large. For example in continuous Blotto player 1 must select an amount of resource to use for each of $|F|$ battlefields, and therefore storing a pure strategy requires storing $|F|$ real numbers, which is easy to do. Thus, Redundant Fictitious Play can be feasibly applied to continuous games, while the classical version cannot.

The only remaining challenge for continuous games is the best response computation, which may be challenging for certain complex utility functions. However, for the common assumptions that the pure strategy spaces are compact and the utility functions are continuous, this optimization is typically feasible to compute.

---

[2] Note that there always exists at least one pure-strategy best response to any mixed strategy.

For the continuous Blotto game, we present optimization formulations for computing player 1 and 2's best response below. Both of these are mixed integer linear programs (with a polynomial number of variables and constraints). Note that we are able to construct efficient best response procedures for this game despite the fact that the utility function is discontinuous.

Player 1's best response function is the following, where $X_q$ is a variable denoting the amount of resources put on slot $q$, and $Y_{t,o,q}$ is the amount of resources put on slot $q$ under outcome $o$ by player 2's fixed strategy at iteration $t$:

Maximize $\sum_t \sum_o \sum_q \left( p(o) \cdot b_{t,o,q} \cdot v_{o(q)} \right)$ subject to:

$$b_{t,o,q} = 1 \rightarrow X_q \geq Y_{t,o,q} + \delta \text{ for all } t, o, q \tag{1}$$

$$\sum_q X_q = B_1 \tag{2}$$

$$0 \leq X_q \leq B_1 \text{ for all } q \tag{3}$$

$$b_{t,o,q} \text{ binary in } \{0, 1\} \text{ for all } t, o, q \tag{4}$$

The constraints in Equation (1) are called indicator constraints and state that if the binary variable $b_{t,o,q}$ has value equal to 1, then the linear constraint $X_q \geq Y_{t,o,q} + \delta$ must hold. Indicator constraints are supported by many integer-linear program optimization solvers, such as CPLEX and Gurobi. We could additionally impose indicator constraints $b_{t,o,q} = 0 \rightarrow X_q \leq Y_{t,o,q}$; however, these are unnecessary and would significantly increase the size of the problem. To see the correctness of the procedure, suppose that $X_q \geq Y_{t,o,q} + \delta$ and $\sum_q X_q = B_1$ but that $b_{t,o,q} = 0$. Then the objective clearly increases by setting $b_{t,o,q} = 1$ instead to include the additional term $p(o) \cdot b_{t,o,q} \cdot v_{o(q)}$. So there cannot exist another solution satisfying the budget and indicator constraints with higher objective value.

While player 1 must assume that the outcome is distributed according to $p$, player 2 is aware of the outcome and therefore can condition their strategy on it. Therefore, player 2 solves a separate optimization for each value of $o \in O$ to compute the best response to the strategy of player 1.

Player 2's best response function given outcome $o \in O$ is the following, where $Y_q$ is a variable denoting the amount of resources put on slot $q$ and $X_{t,q}$ is the amount of resources put on slot $q$ according to player 1's fixed strategy at iteration $t$:

Maximize $\sum_t \sum_q \left( b_{t,q} \cdot v_{o(q)} \right)$ subject to:

$$b_{t,q} = 1 \rightarrow Y_q \geq X_{t,q} \text{ for all } t, q \tag{5}$$

$$\sum_q Y_q = B_2 \tag{6}$$

$$0 \leq Y_q \leq B_2 \text{ for all } q \tag{7}$$

$$b_{t,q} \text{ binary in } \{0, 1\} \text{ for all } t, q \tag{8}$$

Correctness of player 2's best response function follows by similar reasoning to that of player 1's. Player 1's best response optimization has $T'M|Q|$ binary variables $b_{t,o,q}$, where $T'$ is the current algorithm iteration and $M = |O|$ denotes the number of outcomes, and $|Q|$ continuous variables $X_q$. Since the number of indicator constraints is also $T'M|Q|$, the size of the formulation is $O(TM|Q|) = O(TM|F|)$, which is polynomial in all of the input parameters. Similarly, player 2 must solve $M$ optimizations, each one with size $O(T'|Q|)$. Note that in practice this algorithm could be parallelized by solving each of these $M + 1$ optimizations simultaneously on separate cores as opposed to solving them sequentially (in our implementation we solve them sequentially). However, since player 1's optimization is much larger than each of player 2's, the bottleneck step is player 1's optimization, and such a parallelization may not provide a significant reduction in the runtime.

Note that as we run successive iterations of Algorithm 1, the size of these optimization problems becomes larger, since the opponent's strategy is a mixture over $t$ pure strategies,

where $t$ is the current algorithm iteration. We have seen that the number of variables and constraints scales linearly in $t$. Therefore, we expect earlier iterations of the algorithm to run significantly faster than later iterations. We will see the exact magnitude of this disparity in the experiments in Section 4. A potential solution to this issue would be to include an additional parameter $K$ in Algorithm 1. Instead of computing a best response to the mixture over all $t$ of the opponent's pure strategies, a subset of $K$ of them is selected by sampling and a best response is computed just to a uniform mixture over the pure strategies in the sampled subset. This sampling would occur for each iteration, so a potentially different subset of size $K$ would be selected at each iteration. This would ensure that the complexity of the best response computations remains constant over all iterations and does not become intractable for later iterations. This approach would be unbiased and produces the same result in expectation over the sampling outcomes. However, it may lead to high variance in results and lead to poor convergence in practice. Perhaps this could be mitigated by performing multiple runs of the sampling algorithm in parallel and selecting the run with lowest value of $\epsilon$.

Note that Algorithm 1 can be applied to extensive-form imperfect-information games in addition to simultaneous strategic-form games (in fact the continuous Blotto game that we apply it to has imperfect information for player 1, since player 1 does not know the value of $o$ while player 2 does). As long as pure strategies can be represented and best responses can be computed efficiently (which are both the case for imperfect-information games), the algorithm can be applied. Also note that while we presented the algorithm just for a two-player game, it can also be run on multiplayer games (just as for standard fictitious play). The best response computations are still just a single agent optimization problem given fixed strategies for the opposing players. In fact, fictitious play has been demonstrated to obtain successful convergence to Nash equilibrium in a variety of multiplayer settings [35], despite the fact that it is not guaranteed to converge to Nash equilibrium in general for games that are not two-player zero-sum.

We can compute $v_1^*[t]$ and $\epsilon_1[t]$ for Algorithm 1 in the continuous Blotto game using the procedures depicted in Algorithms 2 and 3 (and analogously for $v_2^*[t]$ and $\epsilon_2[t]$).

---

**Algorithm 2** Procedure to compute $v_1^*[t]$ in continuous Blotto

---

$v_1^*[t] \leftarrow 0$
**for** $t_1 = 0$ to $t$ **do**
    **for** $t_2 = 0$ to $t$ **do**
        **for** $o \in O$ **do**
            **for** $q \in Q$ **do**
                **if** $S_1[t_1](q) \geq S_2[t_2](o,q) + \delta$ **then**
                    $v_1^*[t] \leftarrow v_1^*[t] + p(o)v_{o(q)}$
                **else if** $S_1[t_1](q) \leq S_2[t_2](o,q)$ **then**
                    $v_1^*[t] \leftarrow v_1^*[t] - p(o)v_{o(q)}$
$v_1^*[t] \leftarrow \frac{v_1^*[t]}{(t+1)^2}$
**return** $v_1^*[t]$

---

---

**Algorithm 3** Procedure to compute $\epsilon_1[t]$ in continuous Blotto

---

$\quad \epsilon_1[t] \leftarrow 0$
$\quad$ **for** $t_2 = 0$ to $t - 1$ **do**
$\quad\quad$ **for** $o \in O$ **do**
$\quad\quad\quad$ **for** $q \in Q$ **do**
$\quad\quad\quad\quad$ **if** $S_1[t](q) \geq S_2[t_2](o,q) + \delta$ **then**
$\quad\quad\quad\quad\quad \epsilon_1[t] \leftarrow \epsilon_1[t] + p(o)v_{o(q)}$
$\quad\quad\quad\quad$ **else if** $S_1[t](q) \leq S_2[t_2](o,q)$ **then**
$\quad\quad\quad\quad\quad \epsilon_1[t] \leftarrow \epsilon_1[t] - p(o)v_{o(q)}$
$\quad \epsilon_1[t] \leftarrow \frac{\epsilon_1[t]}{t}$
$\quad \epsilon_1[t] \leftarrow \epsilon_1[t] - v_1^*[t-1]$
$\quad$ **return** $\epsilon_1[t]$

---

## 4. Experiments

We experimented on a game with three battlefields $f_1, f_2, f_3$, with values $v_1 = 0.7$, $v_2 = 0.2$, $v_3 = 0.1$, and three outcomes (each with probability $\frac{1}{3}$):

- Outcome 1 has $f_1$ in slot 1, $f_2$ in slot 2, $f_3$ in slot 3.
- Outcome 2 has $f_3$ in slot 1, $f_1$ in slot 2, $f_2$ in slot 3.
- Outcome 3 has $f_2$ in slot 1, $f_3$ in slot 2, $f_1$ in slot 3.

We assume that player 2 observes the outcome while player 1 does not. We used a budget $B_1 = 10$ for player 1 and $B_2 = 7$ for player 2. We used $\delta = 0.0001$. We used the default feasibility tolerance in Gurobi, which is $1.0 \times 10^{-6}$. We ran our algorithm for 5000 iterations and computed $\epsilon_i$ for each player every 10 iterations. Recall that we defined the exploitability of the computed strategies at iteration $t$ as $\epsilon[t] = \max_i \epsilon_i[t]$. The experiments did not use any sampling and computed the best response against the opponent's full mixed strategy at each iteration using the mixed-integer linear programs described in Section 3. We used the parallel version of Gurobi's mixed integer linear programming solver [36] with six cores on a laptop.

The results are shown in Figure 1. It took slightly under 25,000 s (around 6.9 h) to run 5000 iterations of our algorithm. The final strategies had an exploitability of 0.0307 for player 1 and 0.0292 for player 2, indicating that the strategies constitute an $\epsilon$-equilibrium for $\epsilon = 0.0307$. (After 5000 additional iterations $\epsilon$ decreased further to 0.021.) The exploitability values are not monotonically decreasing, and the lowest value in these experiments was actually obtained with $\epsilon = 0.0259$ at iteration 4480. The expected value for player 1 in the final strategies is $-0.10969$. The exploitability fell below 0.05 for the first time after 1759.4 s (29.3 min), obtaining $\epsilon = 0.0494$ on iteration 1400. From the figure we can also see that the runtimes varied for the different iterations, as expected (nearly half of the 5000 iterations were completed in the first 5000 s).

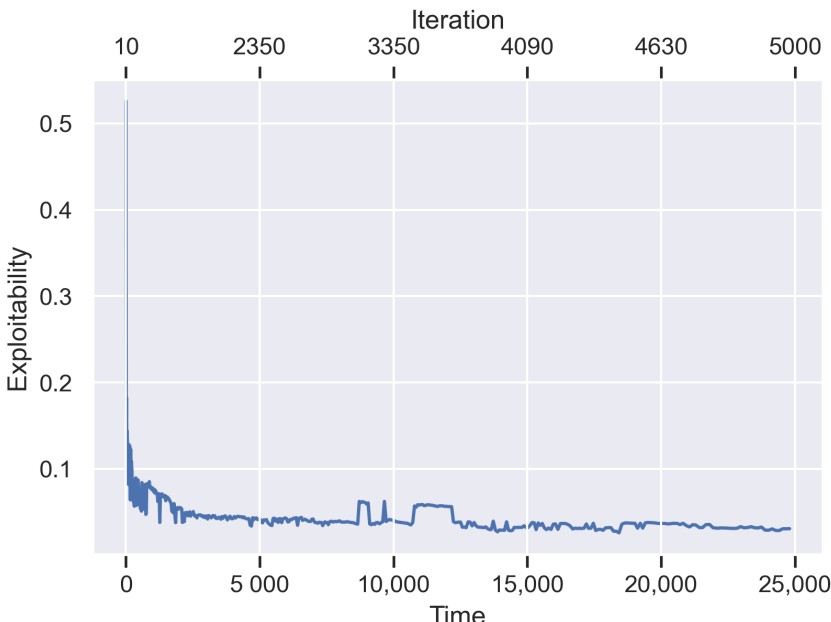

**Figure 1.** Exploitability ($\epsilon$) vs. runtime (seconds) and algorithm iteration for continuous imperfect-information Blotto game.

## 5. Conclusion

We presented a new algorithm for computing Nash equilibrium in a broad class of continuous games. The algorithm is based on integrating a novel variant of fictitious play in which the strategies from all iterations are stored with custom best response functions. Solving continuous games is particularly challenging as a Nash equilibrium is not even guaranteed to exist and mixed strategies may put weight on infinitely many pure strategies; yet for many realistic games it is more natural to model strategies as subsets of real numbers than as integers. We implemented our algorithm on a continuous imperfect-information model of the Blotto game, a well-studied model of resource allocation with applications to national security. We created a new mixed-integer linear program formulation for the best response function. We demonstrated that the algorithm converged quickly to an $\epsilon$-equilibrium for $\epsilon$ equal to 0.03 after 5000 iterations of the algorithm (several hours), which corresponds to 30% of the minimum battlefield value. While the Blotto game has been studied analytically and efficient algorithms have been developed for the discrete case, this is the first algorithm for solving the continuous case.

**Funding:** This research was developed with funding from the Defense Advanced Research Projects Agency (DARPA). The views, opinions and/or findings expressed are those of the author and should not be interpreted as representing the official views or policies of the Department of Defense or the U.S. Government.

**Acknowledgments:** We would like to acknowledge Arctan, Inc., and in particular Peter Dragos, Charles Morefield, Michael Morefield, and Keith Paarporn.

**Conflicts of Interest:** The authors declare no conflict of interest.

1.　Brown, N.; Sandholm, T. Superhuman AI for multiplayer poker. *Science* **2019**, *365*, 885–890. [CrossRef]
2.　Chen, X.; Deng, X. *3-Nash Is PPAD-Complete*; Report No. 134; Electronic Colloquium on Computational Complexity: Rehovot, Israel, 2005; pp. 1–12.
3.　Chen, X.; Deng, X. Settling the Complexity of 2-Player Nash Equilibrium. In Proceedings of the Annual Symposium on Foundations of Computer Science (FOCS), Berkeley, CA, USA, 21–24 October 2006.
4.　Daskalakis, C.; Goldberg, P.; Papadimitriou, C. The Complexity of Computing a Nash Equilibrium. *SIAM J. Comput.* **2009**, *1*, 195–259. [CrossRef]

5.	Ganzfried, S.; Sandholm, T. Computing equilibria in multiplayer stochastic games of imperfect information. In Proceedings of the 21st International Joint Conference on Artificial Intelligence (IJCAI), Pasadena, CA, USA, 11–17 July 2009.
6.	Nash, J. Non-Cooperative Games. Ph.D. Thesis, Princeton University, Princeton, NJ, USA, 1950.
7.	Robinson, J. An iterative method of solving a game. *Ann. Math.* **1951**, *54*, 296–301. [CrossRef]
8.	Gilpin, A.; Peña, J.; Sandholm, T. First-Order Algorithm with $\mathcal{O}(\ln(1/\epsilon))$ Convergence for $\epsilon$-Equilibrium in Two-Person Zero-Sum Games. *Math. Program.* **2012**, *133*, 279–298. [CrossRef]
9.	Kuhn, H.W. Extensive Games and the Problem of Information. In *Contributions to the Theory of Games*; Kuhn, H.W., Tucker, A.W., Eds.; Annals of Mathematics Studies, 28; Princeton University Press: Princeton, NJ, USA, 1953; pp. 193–216.
10.	Archibald, C.; Shoham, Y. Modeling Billiards Games. In Proceedings of the International Conference on Autonomous Agents and Multi-Agent Systems, Budapest, Hungary, 10–15 May 2009.
11.	Glicksberg, I.L. A Further Generalization of the Kakutani Fixed Point Theorem, With Application to Nash Equilibrium Points. *Proc. Am. Math. Soc.* **1952**, *3*, 170–174. [CrossRef]
12.	Fudenberg, D.; Tirole, J. *Game Theory*; MIT Press: Cambridge, MA, USA, 1991.
13.	Stein, N.D.; Ozdaglar, A.; Parillo, P.A. Separable and Low-Rank Continuous Games. *Int. J. Game Theory* **2008**, *37*, 475–504. [CrossRef]
14.	Vorobeychik, Y.; Wellman, M. Stochastic Search Methods for Nash Equilibrium Approximation in Simulation-Based Games. In Proceedings of the International Conference on Autonomous Agents and Multi-Agent Systems, Estoril, Portugal, 12–16 May 2008.
15.	Singh, S.P.; Soni, V.; Wellman, M.P. Computing approximate Bayes-Nash equilibria in tree-games of incomplete information. In Proceedings of the ACM Conference on Electronic Commerce (ACM-EC), New York, NY, USA, 17–20 May 2004.
16.	Ganzfried, S.; Sandholm, T. Computing equilibria by incorporating qualitative models. In Proceedings of the International Conference on Autonomous Agents and Multi-Agent Systems, Toronto, ON, Canada, 10–14 May 2010.
17.	Borel, É. La théorie du jeu et les équations intégrales á noyau symétrique. *C. R. L'Académie* **1921**, *173*, 97–100.
18.	Wikipedia Contributors. Blotto Game—Wikipedia, The Free Encyclopedia. 2020. Available online: https://en.wikipedia.org/wiki/Blotto_game (accessed on 15 May 2020).
19.	Borel, É.; Ville, J. *Applications de la Théorie des Probabilités aux Jeux de Hasard*; Gauthier-Vilars: Paris, France, 1938.
20.	Gross, O.A.; Wagner, R.A. *A Continuous Colonel Blotto Game*; RM-098; Rand Project Air Force: Santa Monica CA, USA, 1950.
21.	Roberson, B. The Colonel Blotto Game. *Econ. Theory* **2006**, *29*, 1–24. [CrossRef]
22.	Hart, S. Discrete Colonel Blotto and General Lotto Games. *Int. J. Game Theory* **2008**, *36*, 441–460. [CrossRef]
23.	Ahmadinejad, A.; Dehghani, S.; Hajiaghayi, M.; Lucier, B.; Mahini, H.; Seddighin, S. From Duels to Battlefields: Computing Equilibria of Blotto and Other Games. In Proceedings of the AAAI Conference on Artificial Intelligence, Phoenix, AZ, USA, 12–17 February 2016.
24.	Behnezhad, S.; Dehghani, S.; Derakhshan, M.; Hajiaghayi, M.T.; Seddighin, S. Faster and Simpler Algorithm for Optimal Strategies of Blotto Game. In Proceedings of the AAAI Conference on Artificial Intelligence (AAAI), San Francisco, CA, USA, 4–9 February 2017.
25.	Adamo, T.; Matros, A. A Blotto Game with Incomplete Information. *Econ. Lett.* **2009**, *105*, 100–102. [CrossRef]
26.	Kovenock, D.; Roberson, B. A Blotto game with multi-dimensional incomplete information. *Econ. Lett.* **2011**, *113*, 273–275. [CrossRef]
27.	Paarporn, K.; Chandan, R.; Alizadeh, M.; Marden, J.R. Characterizing the interplay between information and strength in Blotto games. In Proceedings of the 2019 IEEE 58th Conference on Decision and Control (CDC), Nice, France, 11–13 December 2019; pp. 5977–5982.
28.	Chandan, R.; Paarporn, K.; Marden, J.R. When showing your hand pays off: Announcing strategic intentions in Colonel Blotto games. In Proceedings of the American Control Conference, Denver, CO, USA, 1–3 July 2020.
29.	Brown, G.W. Iterative Solutions of Games by Fictitious Play. In *Activity Analysis of Production and Allocation*; Koopmans, T.C., Ed.; John Wiley & Sons: Hoboken, NJ, USA, 1951; pp. 374–376.
30.	Fudenberg, D.; Levine, D. *The Theory of Learning in Games*; MIT Press: Cambridge, MA, USA, 1998.
31.	Ganzfried, S.; Sandholm, T. Computing an approximate jam/fold equilibrium for 3-player no-limit Texas hold 'em tournaments. In Proceedings of the International Conference on Autonomous Agents and Multi-Agent Systems, Estoril, Portugal, 12–16 May 2008.
32.	Rabinovich, Z.; Gerding, E.; Polukarov, M.; Jennings, N.R. Generalised fictitious play for a continuum of anonymous players. In Proceedings of the 21st International Joint Conference on Artificial Intelligence (IJCAI), Pasadena, CA, USA, 11–17 July 2009.
33.	Rabinovich, Z.; Naroditskiy, V.; Gerding, E.H.; Jennings, N.R. Computing pure Bayesian-Nash equilibria in games with finite actions and continuous types. *Artif. Intell.* **2013**, *195*, 106–139. [CrossRef]
34.	Ganzfried, S.; Laughlin, C.; Morefield, C. Parallel Algorithm for Nash Equilibrium in Multiplayer Stochastic Games with Application to Naval Strategic Planning. In *Lecture Notes in Computer Science, Proceedings of the International Conference on Distributed Artificial Intelligence (DAI), Nanjing, China, 24–27 October 2020*; Springer: Cham, Switzerland, 2020.
35.	Ganzfried, S. Fictitious Play Outperforms Counterfactual Regret Minimization. *arXiv* **2020**, arXiv:2001.11165.
36.	Gurobi Optimization, L. *Gurobi Optimizer Reference Manual*; Gurobi Optimization: Houston, TX, USA, 2019.
