# Peer review of "Algorithm for Computing Approximate Nash Equilibrium in Continuous Games with Application to Continuous Blotto"

_games, doi:10.3390/g12020047_

Round 1

Reviewer 1 Report

This manuscript proposes a new algorithm for computing an approximate Nash equilibrium in continuous games, and shows extensive experimental results in case that the proposed algorithm is applied in a well-known, continuous Blotto games. This manuscript is well-written overall with a comprehensive survey of the literature. Especially, research contributions and limitations of the proposed algorithm (i.e., a variant of fictitious play) are clearly described and the reviewer agree with them.

However, the following minor comments could be reflected for further improvements.

 - In abstract, it is better to add 'approximate' in 'We present a new algorithm for computing Nash equilibrium strategies in continuous games.', although it is clear from the title and the last sentence of abstract.

 - Considering a wider readership of this journal, it is better to specify the full name of PPAD, although such term is widely used in computer science.

 - In line 188 and 189, it is better to add either commas or whitespace for better readability.

Author Response

Thank you for the comments. We have modified the paper according to the suggestions in the new version.

Reviewer 2 Report

This paper addresses the challenging class of problems of continuous games with asymmetric-incomplete information. The approach is based upon extending the fictitious play algorithm, which has been successfully applied to simpler classes of games. Traditional fictitious play is based upon updating each player's mixed strategy in each iteration by mixing in that player's current best response. The issue, when applying the idea to continuous games, is that it can be difficult to represent a mixed strategy over a continuum of pure strategies. In theory, the mixed strategy is a probability distribution function whose domain is the compact set of pure strategies. The algorithm is demonstrated on the continuous Blotto game where one player knows the ordering of the battlefields, while the other does not. For the example shown, the algorithm converges to an epsilon-equilibrium with relatively small epsilon.

Major Comments:
1) The notion of "outcome" and "battlefield ordering" should be unified and better explained. It is clear that one player knows the ordering and one does not, but it is totally unclear as to how this affects the game. Throughout most of the paper, it appears as if the value of winning each battlefield is static (i.e., independent of battlefield order). Therefore, how does Player 2's knowledge of the battlefield order help him? In one place (between lines 290-291) it appears that the battlefield value depends on the outcome, which seems much more sensical. Player 2, knowing the exact value of each battlefield can make better resource allocation decisions compared to Player 1 who must consider all possible outcomes with their respective probability.
2) Overall it is somewhat unclear as to what can be stated about the algorithm. For example, the authors state that existence of a Nash equilibrium for this problem is not guaranteed. What might happen if the algorithm was applied to a problem without a Nash equilibrium? If we could guarantee existence for a particular problem, then it would be nice to have a result which says that convergence is guaranteed. The discussion about the complexity of computing best responses is nice, but the analysis of the algorithm as a whole is somewhat lacking.

Minor Comments:
1) Please see the attached PDF.

Author Response

I see how the wording about the ordering is confusing in the paper, and I have clarified it throughout the paper. Both players select a strategy for selecting each slot, not for each battlefield (there are the same number of battlefields as slots). Player 1 does not know which battlefield will be in each slot, only a probability distribution over orderings, while player 2 knows the exact ordering. For example, suppose there is probability 50% that the ordering is (f1, f2, f3), and probability 50% that the ordering is (f3 f2 f1). If player 1 puts an amount of resources x1 on the first slot, then there is 50% chance that these resources will be for f1, and 50% that they will be for f3 (depending on whether ordering 1 or ordering 2 is selected). This is the same model as has been used for prior research on the version of the problem with asymmetric imperfect information, which is cited in the paper.

It would be nice to be able to prove a theoretical result about the algorithm, but that is beyond the scope of this paper (and it is possible that no general result can be proven). Fictitious play is guaranteed to converge to Nash equilibrium in two-player zero-sum games, and certain other classes of 2-player games (see the Wikipedia page for a list). For games not on the list, the algorithm can still be run even though convergence is not guaranteed. It can then be checked relatively easily how small the epsilon is for the computed strategies (maximum amount that a player can gain by deviating), and it is possible that we are “lucky.” While some counterexamples have been constructed for which it does not converge, it has been shown to converge empirically in several classes of games with more than two players (cited in the paper). So the best we can really say at this time is that the algorithm can be run efficiently and the quality of convergence must be verified at the end. I think that it is possible that convergence can be proven for a broad class of two-player zero-sum continuous games, but such a proof will likely be very challenging and technical, and is something we can look at in the future.

If the game (or game class) is not guaranteed to have a Nash equilibrium, then of course convergence to one cannot be guaranteed. But similarly we can still run the algorithm and empirically verify the value of epsilon in the computed strategies.

Thank you for the detailed comments in the pdf file, I have addressed them and highlighted the changes.